# Soluble CD14 in Breast Milk and Its Relation to Atopic Manifestations in Early Infancy

**DOI:** 10.3390/nu11092118

**Published:** 2019-09-05

**Authors:** Bahrul Fikri, Yumi Tani, Kazue Nagai, Masumi Sahara, Chisako Mitsuishi, Yaei Togawa, Taiji Nakano, Fumiya Yamaide, Hiroshi Ohno, Naoki Shimojo

**Affiliations:** 1Department of Pediatrics, Graduate School of Medicine, Chiba University, Chiba 2608670, Japan; 2Department of Pediatrics, Japanese Red Cross Katsushika Maternity Hospital, Tokyo 1240012, Japan; 3Research and Education Center of Health Sciences, Gunma University Graduate School of Health Sciences, Gunma 3718510, Japan; 4Department of Dermatology, Graduate School of Medicine, Chiba University, Chiba 2608670, Japan; 5RIKEN Center for Integrative Medical Sciences, Yokohama City, Kanagawa 2300045, Japan

**Keywords:** soluble CD14, breast milk, atopic dermatitis, allergen sensitization

## Abstract

Soluble CD14 (sCD14) is one of the immunomodulatory factors in breast milk (BM). Although it may be involved in the prevention of atopic symptoms and sensitization to both food and inhalant allergens, conflicting evidence exists concerning its protective effects. In this study, we investigated the relationship between sCD14 in colostrum and 1-month BM, and the development of atopic dermatitis (AD) and sensitization to food and aeroallergens at 9 months of age in infants who were exclusively or almost exclusively breastfed up to 4 months of age. BM samples were collected from lactating mothers who participated in a 2 × 2 factorial, randomized, nontreatment controlled trial study set in Tokyo, which looked at the efficacy of emollients and synbiotics in preventing AD and food allergy in children during the first year of life. A total of 258 colostrum samples and 269 1-month BM samples were analyzed. We found that one-month BM sCD14 levels in the AD group were significantly lower than in the non-AD group. Levels of sCD14 in 1-month BM were not related to allergen sensitization in the overall analysis, but egg white sensitization correlated inversely with 1-month BM sCD14 in infants without AD. The results suggest that sCD14 in BM may be involved in atopic manifestations in early infancy.

## 1. Introduction

Exclusive breastfeeding for the first 4–6 months of life is endorsed for primary prevention of allergic diseases in children on the basis of epidemiological findings and expert opinion [1]. The protective effect of breast milk (BM) against atopic dermatitis (AD) was shown by a meta-analysis in 2001 [2]. However, a later meta-analysis in 2009 indicated no relation between feeding methods and AD [3]. Furthermore, a recent large-scale epidemiological study in Japan showed an apparent risk of developing AD due to breastfeeding [4]. One of the reasons for these differences in results with time may be changes in environmental factors in modern society, including immune modulators in breast milk [5]. 

Human BM contains a number of specific and nonspecific anti-inflammatory substances. Soluble CD14 (sCD14) is one of the BM components that aid neonatal gut function and development [6,7,8], modulates immune function, and regulates inflammation [9,10,11,12]. It acts as a co-receptor along with toll-like receptor 4 (TLR4) and MD-2 in the detection of LPS and binds to LPS more efficiently in the presence of lipopolysaccharide-binding protein [13,14,15]. An interaction between polymorphism in the *CD14* gene and endotoxin exposure may play a role in the development of allergies [16].

Consensus on the involvement of human BM sCD14 in the development of atopic manifestations in childhood has not been established. Several cohort studies reported that lower sCD14 in BM were associated with allergen sensitization or allergic symptoms [17,18,19,20,21]. By contrast, Savilahi et al. documented that higher sCD14 levels in BM 3 months postpartum were associated with IgE-mediated allergic disorders by the age of 5 years [22]. Other studies found sCD14 levels either in colostrum or mature milk were not associated with subsequent development of eczema or atopic sensitization during early life [23,24,25,26]. The substantial variations in study protocols, such as time of outcome measurement, selection of subjects, change in other immunomodulators in BM with time, etc., may contribute to these conflicting results. 

In this study, to understand the involvement of sCD14 in BM in atopic manifestations in early infancy, we studied the association of sCD14 levels in colostrum and 1-month BM, with the development of AD and allergen sensitization at 9 months of age in exclusively breastfed or almost exclusively breastfed infants in a birth cohort in Japan.

## 2. Materials and Methods

### 2.1. Participants

The present study used the findings of our previous study (Katsushika study, a 2 × 2 factorial, randomized, nontreatment-controlled trial study to evaluate the prevention of AD or food allergy (FA) by use of skincare and synbiotics in infants) (*n* = 549). The study was approved by the Chiba University Ethics Committee (reference no. 2067) and was registered at the University Hospital Medical Information Network (JPRN-UMIN000010838). There was no significant preventive effect of skincare and/or synbiotic on the development of AD up to 1 year of age [27] Children who were exclusively or almost exclusively breastfed up to four months were studied in the present study. Feeding method up to 4 months was determined from a questionnaire completed by mothers when infants were 6 months of age.

### 2.2. Definition of Maternal Allergic History

A positive maternal allergic history was defined as, physician-diagnosed asthma, AD, rhinoconjunctivitis, or FA in the mother, as declared in a self-reported questionnaire. 

### 2.3. Definition of Feeding Method

Exclusively or almost exclusively breastfed infant was defined as one who had no nutrition other than breast milk or mostly breast milk nutrition until 4 months of age.

### 2.4. Definition of Infant’s AD and Infant’s Allergic Sensitization

Infants were examined for AD by a pediatrician at 9 months of age, and the presence or absence of AD was determined using guidelines of the Japanese Dermatological Association [28]. Specific IgE for hen’s egg white, cow’s milk, house dust mite, and cat dander were also measured at 9 months of age by ImmunoCap^®^ system (Phadia KK., Tokyo, Japan). Infants were considered to be sensitized if specific serum IgE levels were ≥0.7 kU/L against one or more of the tested foods or inhalant allergens.

### 2.5. Colostrum and BM Collection

BM samples were collected from mothers, 3–5 days postpartum, and 1 month postpartum. Each sample was frozen at −80 °C within 12 h after collection. For analysis of sCD14 in this study, frozen BM samples were thawed and centrifuged at 10,000 rpm for 10 min. The cellular debris and fat layer were discarded, and the aqueous fraction was centrifuged again at 10,000 rpm for 5 min. The resulting aqueous layer was used for analysis.

### 2.6. Measurement of sCD14

The concentration of sCD14 in the aqueous phase of BM samples was measured in duplicate using a commercial human sCD14 ELISA kit according to the manufacturer’s protocol (Human sCD14 DuoSet ELISA kit, R & D System, Minneapolis, MN, USA). Samples were assayed at a dilution of 1:40,000 for colostrum, and 1:10,000 for 1-month BM. Recombinant human sCD14 was used as a positive control and the reagent diluent buffer provided by the manufacturer was used as a negative control. Seven-point standard curves using 1:2 serial dilutions were generated in the range of 4000–62.5 pg/mL. sCD14 concentration was determined based on the standard curve generated by PLATEmanager V5 for Sunrise program (Wako Pure Chemical Industries, Ltd., Osaka, Japan) using a four-parameter equation and results were reported in μg/mL. 

### 2.7. Statistical Analysis

Data were analyzed as continuous or dichotomous data (detected or not detected). Results for categorical variables, including maternal allergic history, mode of delivery, gender, season of birth, and age of mother were compared with the outcomes by Chi-square test. Wilcoxon rank sum test or Kruskal–Wallis test for nonparametric analysis was used to compare sCD14 concentration in colostrum and 1-month BM with the outcomes. To assess the possible association between colostral sCD14 or 1-month BM sCD14 levels and outcomes, potential confounding factors (mode of delivery, maternal allergic history, gender, season of birth, and age of mother) were included in a nominal logistic regression model. The dose-dependent trend of BM sCD14 levels with respect to the outcomes was analyzed by the Cochran–Armitage trend test. A *p*-value < 0.05 was considered significant. Statistical analysis was performed using JMP13 (SAS Institute Inc., Cary, NC, USA).

## 3. Results

### 3.1. BM Samples and sCD14 Concentration

A total of 258 colostrum samples and 269 1-month BM samples were collected, respectively. Two hundred and forty-six of colostrum and 1-month BM were paired samples. The characteristics of the samples are summarized in Table 1. There was a significant change in sCD14 levels between the two time points; colostrum (median 20.8 μg/mL, range 2.2–45.9 μg/mL) and 1-month BM (median 7.87 μg/mL, range 0.88–28.3 μg/mL) (Wilcoxon rank sum test, *p* < 0.0001) (Figure 1a). We found no significant difference in sCD14 levels in colostrum (*p* = 0.14, Wilcoxon rank sum test) and 1-month BM (*p* = 0.82, Wilcoxon rank sum test) between mothers with or without allergic history (Figure 1b).

### 3.2. Association between BM sCD14 Levels and AD at 9 Months of Age

Colostral sCD14 levels were not different between children who developed AD at 9 months of age and those who did not (*p* = 0.804) (Table 2). In contrast, 1-month BM sCD14 levels in the AD group were significantly lower than those in the non-AD group (*p* = 0.001). This association was confirmed by using logistic regression with adjustment for maternal allergic history, gender, mode of delivery, season of birth, and age of mother (adjusted *p* = 0.002). In order to determine the effect size of 1-month BM sCD14 on the development of AD, we divided 1-month BM sCD14 levels into four groups and analyzed by logistic regression with adjustment for the confounders, showing that higher levels of 1-month BM sCD14 significantly decreased the risk of the development of AD at 9 months (Table 3). 

### 3.3. Association between BM sCD14 Levels and Sensitization to Allergens at 9 Months of Age

We analyzed the association between colostral sCD14 or 1-month BM sCD14 levels, and sensitization to egg white (EWS), cow’s milk, house dust mite, and cat dander. The level of sCD14 in 1-month BM was significantly lower in the EWS group compared to the control group (*p* = 0.01) (Table 4). However, this association was lost after adjustment for other confounding factors (*p* = 0.183). Since AD had a strong effect on sensitization, we stratified subjects based on the presence and absence of AD. EWS was significantly decreased by increasing 1-month BM sCD14 levels in non-AD infants but not in AD infants (Figure 2) (*p* = 0.042, Cochran–Armitage trend test). 

## 4. Discussion

In the present study, we found that sCD14 levels in 1-month BM were significantly lower in children with AD compared to those without AD at 9 months of age who were exclusively or almost exclusively breastfed up to 4 months. Our findings are in accord with Savilahti and co-workers’ results, which showed that atopic symptoms and IgE sensitization at 4 years of age were associated with lower colostral sCD14 levels [18]. Recently, Hua et al. reported that lower sCD14 levels in the colostrum and two-month BM were associated with AD at 2 years of age in children who were born to mothers with allergies in univariate analysis. Unfortunately, the association was lost while performing a logistic regression analysis. Notably, in our study, the association between 1-month BM sCD14 levels and nine-month AD were independent of maternal allergic history and gender, which were also associated with nine-month AD. In contrast to the present study, Ismail et al. reported that sCD14 levels in 28-day BM were not associated with the development of eczema or allergen sensitization in infants at 12 months [24]. The reason for the discrepancy is not clear. However, there is a possibility that sCD14 in BM was modified since mothers took probiotics during pregnancy in their study [19]. In our study, synbiotics were given to the babies but not the mothers; thus, this intervention may not have an effect on the mother’s status. We included the intervention in logistic regression analysis and the results were not changed (data not shown). This analysis of results indicated that there was no synergistic effect of the treatments and 1-month BM sCD14 in the development of AD at 9 months. One-month BM sCD14 levels were independently associated with the outcome. Another reason for this discrepancy might be different characteristics of the enrolled subjects. Our study subjects were from an unselected population while the aforementioned study recruited high-risk children with a family history of allergy. In fact, the prevalence of AD in our study is lower than that in their study, and EASI score in children with AD in our study was not high (data not shown).

We found lower 1-month BM sCD14 levels were associated with the development of EWS at 9 months of age in non-AD infants. Our results are similar to other studies that found an association between BM sCD14 with allergen sensitization in childhood [18,22]. However, the effect of sCD14 on allergen sensitization seems to be much weaker because its effect is not observed in the presence of AD, which is a strong enhancing factor for allergen sensitization. 

Our study has some limitations. The outcomes were set at 9 months of age in this study to minimize factors other than breastfeeding. Longer follow-up is necessary to determine the long-term effect of sCD14 in BM. Our study did not look at the role of sCD14 in BM in the development of AD in the high-risk population, which might be important for personalized prevention. 

In conclusion, we found that low sCD14 levels in 1-month BM were significantly associated with the development of AD at 9 months of age and weakly associated with sensitization to egg white at 9 months in non-AD infants. The results suggest that the sCD14 in BM may be a protective factor for subsequent allergic diseases and/or allergen sensitization. 

## Figures and Tables

**Figure 1 nutrients-11-02118-f001:**
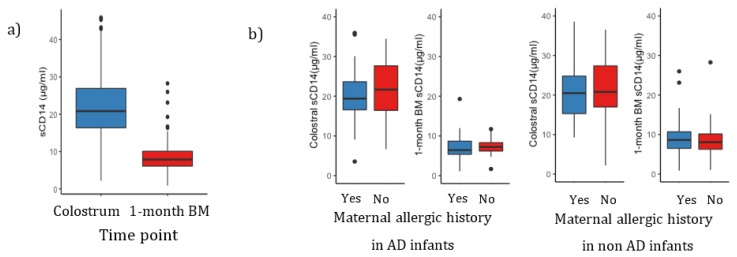
(**a**) Levels of sCD14 in colostrum (*n* = 258) and in 1-month BM (*n* = 269). There was a significant difference between colostral sCD14 and 1-month BM sCD14 levels (Wilcoxon rank sum test, *p* < 0.0001) and (**b**) Colostral sCD14 and 1-month BM sCD14 levels with or without maternal allergic history. In AD or non-AD infants, sCD14 levels in BM were not significantly different between mothers with allergic history and those without allergic history (Wilcoxon rank sum test, *p* > 0.05). The levels are shown as median with interquartile ranges.

**Figure 2 nutrients-11-02118-f002:**
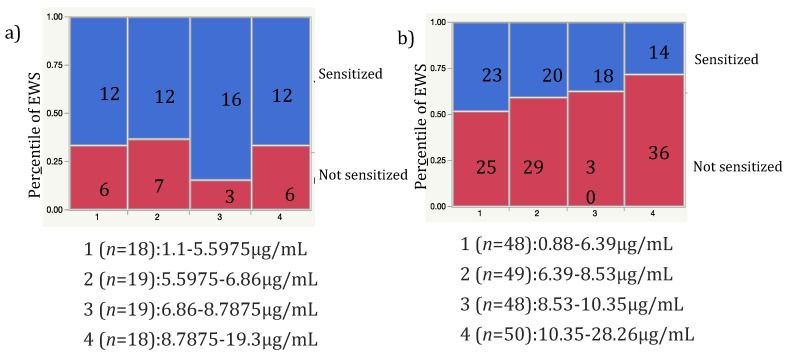
Dose-dependent trend for sCD14 levels in 1-month BM to EWS at 9 months of age in AD (**a**) and non-AD (**b**) infants. EWS decreased significantly with increasing sCD14 levels in 1-month BM in non-AD infants (Cochran–Armitage trend test, *p* = 0.042).

**Table 1 nutrients-11-02118-t001:** Characteristics of colostrum and 1-month BM samples.

Characteristics	Colostrum	1-Month BM
	*n* = 258 *	*n* = 269 *
Maternal allergic history, *n* (%)		
Yes	123 (48)	125 (46)
No	135 (52)	144 (54)
Sex, *n* (%)		
Male	132 (51)	138 (51)
Female	126 (49)	131 (49)
Mode of delivery, *n* (%)		
Vaginal delivery	203 (79)	213 (79)
Caesarian delivery	55 (21)	56 (21)
Season of birth, *n* (%)		
Spring	82 (32)	83 (31)
Summer	72 (28)	71 (26)
Autumn	47 (18)	55 (21)
Winter	57 (22)	60 (22)
Age of mother (year), *n* (%)		
≤25	5 (2)	5 (2)
>25–35	167 (65)	173 (64)
>35	86 (33)	91 (34)
9-month follow-up		
Atopic dermatitis, *n* (%)		
Yes	69 (27)	74(27.5)
No	184 (71)	195 (72.5)
Lost to follow-up	5 (2)	0 (0)
Egg white sensitization, *n* (%)		
Yes	121 (47)	127 (47)
No	135 (52)	142 (53)
Lost to follow-up	2 (1)	0 (0)
Cow’s milk sensitization, *n* (%)		
Yes	23 (9)	26 (10)
No	232 (90)	242 (89)
Lost to follow-up	3 (1)	1 (1)
House dust mite sensitization, *n* (%)		
Yes	11 (4)	13 (5)
No	245 (95)	256 (95)
Lost to follow-up	2 (1)	0
Cat sensitization, *n* (%)		
Yes	8 (3)	7 (2)
No	247 (96)	261 (97)
Lost to follow-up	3 (1)	1 (1)

* Two hundred and forty-six of colostrum and 1-month BM were paired samples.

**Table 2 nutrients-11-02118-t002:** Association of BM sCD14 with atopic dermatitis (AD) at 9 months of age.

BM sCD14	AD	Non-AD	*p*-Value *	Adjusted *p*-Value ^†^
Colostrum **(μg/mL)**	20.87 (3.55–45.31)	20.91 (2.2–45.9)	0.804	0.971
1-month BM **(μg/mL)**	6.86 (1.1–19.3)	8.53 (0.88–28.26)	0.001	0.002

Levels are shown as median with interquartile range. * Groups were compared by Wilcoxon rank sum test. ^†^ Logistic regression analysis adjusted for gender, mode of delivery, maternal allergic history, season of birth, and age of mother.

**Table 3 nutrients-11-02118-t003:** Logistic regression analysis for the effects of 1-month BM sCD14 on AD at 9 months of age.

BM sCD14	sCD14 Levels (μg/mL)	Adjusted OR (95% CI)	*p*-Value
Very low	0.88–6.104	1	
Low	6.105–7.86	0.78 (0.36–1.68)	0.529
Middle	7.87–10.10	0.36 (0.15–0.89)	0.028
High	10.11–28.26	0.38 (0.16–0.89)	0.027

Logistic regression analysis adjusted for gender, mode of delivery, maternal allergic history, season of birth, and age of mother. OR (95% CI): odds ratio (95% confidence interval).

**Table 4 nutrients-11-02118-t004:** Association of BM sCD14 with sensitization to egg white (EWS) at 9 months of age.

BM sCD14	EWS	Non-EWS	*p*-Value *	Adjusted *p*-Value ^†^
Colostrum **(μg/mL)**	20.87 (8.82–45.9)	20.78 (2.2–45.7)	0.777	0.871
1-month BM **(μg/mL)**	7.26 (1.05–16.71)	8.57 (0.88–28.26)	0.01	0.183

Levels are shown as median with interquartile range. * Groups were compared by Wilcoxon rank sum test. ^†^ Logistic regression analysis adjusted for gender, mode of delivery, maternal allergic history, season of birth, age of mother, and AD at 9 months.

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
