# Peer review of "Soluble CD14 in Breast Milk and Its Relation to Atopic Manifestations in Early Infancy"

_nutrients, 2019, doi:10.3390/nu11092118_

Round 1

Reviewer 1 Report

The aim of this article of Fikri et al. is to study the association of the sCD14 in colostrum or 1-month breast-milk and the atopic manifestations at 9 month of age.

Even if it’s interesting, this article requires major revisions.

Major comments:

- The article does not appear with strong novelty compared to other studies already published. The authors must put forward what is new in their study.

- It is indicated that the children who participated in this present study received synbiotics. The authors mention that this has no effect on mother’s status, but it can have an effect on babies. Their previous study did not show any significant preventive effect of symbiotic alone on the development of AD up to 1 year of age. However, a synergistic effect between sCD14 levels in milk and synbiotics ingested by infants on the development of atopic manifestations could be considered. I suggest to the authors to look at a possible association. Making the link between sCD14 levels, synbiotics intake and atopic events would be very innovative.

- How do you explain that you do not find the same results with colostrum?

Minor comments:

- Table 1 is not clear. What do the authors want to highlight?

- How do you explain that there is 5 “lost to follow up” for colostrum while there is 0 for BM? Does it mean that they are not the same mothers? In this case, it is better to explain it.

- The age of the mothers is a data which would be interesting to know because it can have an influence on the level of sCD14 in the milk.

- Form of the data must be indicated in the legend of the figures. What is the type of boxplot used?

- Figure 1b: Putting the 4 graphs at the same scale would simplify reading.

Author Response

Dear Reviewer,

Thank you for the extensive review and the valuable comments. We revised the manuscript according to the reviewers’ comments and re-submit our manuscript entitled “Soluble CD14 in breast milk and its relation to atopic manifestation in early infancy” (Manuscript ID: nutrients-561909). We provide the pdf file (in the attachment) with point-by-point responses. In addition, to improve our paper quality, we changed and added table 1,2,3, and 4. We also added the explanation about the definition of feeding method (Section 2.3. in Material and Methods in revised MS) to make them clear.

We would like to thank you for your helpful comments, and we believe that the revised manuscript is more balanced and improved. We hope that the revised manuscript will be acceptable for publication in Nutrients.

Point 1:
The article does not appear with strong novelty compared to other studies already published.
The authors must put forward what is new in their study.
Response 1:
Thank you for the comment. So far, there has been no consensus on the involvement of
sCD14 in the development of allergic diseases and allergen sensitization. We thought several
possibilities such as feeding methods, study population, and its size, the involvement of
factors affecting allergic outcomes and timing of evaluation of outcomes might be
responsible for conflicting results. In this study, we recruited unselected population with
substantial numbers, got precise and accurate feeding methods and duration to select non
formula-fed (breastfed) children, and focused on AD and sensitization in early infancy (9
months) to decrease the effect of factors other than breast milk as much as possible. The
results indicated that sCD14 in 1M BM may be associated with the development of AD. In
addition, sCD14 in 1M BM might have some effect in food allergen sensitization in early
infancy in non-AD children. I believe our data is valuable and important for future metaanalysis
to draw more reliable information about the relation between BM sCD14 and
allergic outcomes and intervention.
Point 2:
It is indicated that the children who participated in this present study received synbiotics. The
authors mention that this has no effect on mother's status, but it can have an effect on babies.
Their previous study did not show any significant preventive effect of symbiotic alone on the
development of AD up to 1 year of age. However, a synergistic effect between sCD14 levels
in milk and synbiotics ingested by infants on the development of atopic manifestations could
be considered. I suggest to the authors to look at a possible association. Making the link
between sCD14 levels, synbiotics intake and atopic events would be very innovative.
Response 2:
Thank you for the important point. We have analyzed the possible effects of treatment
(including synbiotic) to the babies and the development of AD and EWS at 9 months of age
by multivariate analysis. We did not find a synergistic effect and significant association as to
AD or EWS at 9 months.
Point 3:
How do you explain that you do not find the same results with colostrum?
Response 3:
Thank you for the question. Levels of colostrum varied day by day. Its great variance is
shown in Figure 1. In contrast, 1-month BM was more uniform and less variable. This might
be one of the reasons why we did not observe association.
Point 4:
Table 1 is not clear. What do the authors want to highlight?
Response 4:
Thank you for the comment. Table 1 is the description of basic data of the study and we think
this data is necessary for readers.
Point 5:
How do you explain that there is 5 “lost to follow up” for colostrum while there is 0 for BM?
Does it mean that they are not the same mothers? In this case, it is better to explain it.
Response 5:
Thank you for the comment. Lost to follow up for colostrum and 1 month BM are not same
infants. Only a few samples are not from the same mothers.
Point 6:
The age of the mothers is a data which would be interesting to know because it can have an
influence on the level of sCD14 in the milk.
Response 6:
Thank you for the advice. We added the mother's age in table 1. There was no significant
association between age of mothers and BM sCD14 in Colostrum or 1-months BM. We
included the age of the mother as a confounding factor in all multivariate analysis.
Point 7:
Form of the data must be indicated in the legend of the figures. What is the type of boxplot
used?
Response 7:
Thank you for the comments. We added the explanation of the data in the figure legend
(including the type of the boxplot).
Point 8:
Figure 1b: Putting the 4 graphs at the same scale would simplify reading.
Response 8:
Thank you for the suggestion. We changed the scale of Fig.1. We also added an explanation
of the data in a figure legend.

Sincerely,

Bahrul Fikri

Department of Peditrics, Graduate School of Medicine

Chiba University, 1-8-1 Inohana, Chuo-ku,

Chiba, 260-8670, Japan.

Reviewer 2 Report

This study provides further evidence that sCD14 levels in breast milk are associated with AD development. The study was reasonably well conducted and written but there are some concerns and points that could be improved. 

- Fig 1a, what is the p value?, and also for 1a and 1b what is being presented (median with box & whiskers?) ?

-Section 3.2 - why not show the data comparing 1 month sCD14 (and also in colostrum) in the AD and non-AD outcomes? Because this represents the major finding of the paper it would be nice to see this in addition to the odds ratios

- Table 2 indicates that colostral sCD14 and BM sCD14 were used in the multiple variable model. The assumption is that one was used for one analysis and then flip-flopped for the other. In addition to being confusing it is not clear that these are important for the analysis

-  percentile is misspelled on the y-axis of Fig 2

-the 1st sentence of the discussion (line 156-158) would benefit from some revision.

Author Response

Dear Reviewers,

Thank you for the extensive review and the valuable comments. We revised the manuscript according to the reviewers’ comments and re-submit our manuscript entitled “Soluble CD14 in breast milk and its relation to atopic manifestation in early infancy” (Manuscript ID: nutrients-561909). In addition, to improve our paper quality, we changed and added table 1,2,3, and 4. We also added the explanation about the definition of feeding method (Section 2.3. in Material and Methods in revised MS) to make them clear.

We would like to thank you for your helpful comments, and we believe that the revised manuscript is more balanced and improved. We hope that the revised manuscript will be acceptable for publication in Nutrients.

Point 1:
Fig 1a, what is the p value?, and also for 1a and 1b what is being presented (median with box
& whiskers?) ?
Response 1:
Thank you for the comments. We added p value in the legend of Figure 1. Data on the
boxplots were shown as median with inter-quartile range.
Point 2:
Section 3.2 - why not show the data comparing 1 month sCD14 (and also in colostrum) in the
AD and non-AD outcomes? Because this represents the major finding of the paper it would
be nice to see this in addition to the odds ratios.
Response 2:
Thank you for your valuable comments. According to the reviewer's suggestion, we
extensively revised and made new tables (Table 2, 3, and 4). It seems that higher sCD14 in
BM may decrease the prevalence of AD to 1/3.
Point 3:
Table 2 indicates that colostral sCD14 and BM sCD14 were used in the multiple variable
model. The assumption is that one was used for one analysis and then flip-flopped for the
other. In addition to being confusing it is not clear that these are important for the analysis
Response 3:
Thank you for the suggestion. We have revised table 2 and table 4.
Point 4:
percentile is misspelled on the y-axis of Fig 2
Response 4:
Thank you for the comment. It was corrected.
Point 5:
the 1st sentence of the discussion (line 156-158) would benefit from some revision.
Response 5:
Thank you, it was revised to be more understandable.

Sincerely,

Bahrul Fikri

Department of Peditrics, Graduate School of Medicine

Chiba University, 1-8-1 Inohana, Chuo-ku,

Chiba, 260-8670, Japan.
